# Gut Microbiota and Neurodevelopment in Preterm Infants: Mechanistic Insights and Prospects for Clinical Translation

**DOI:** 10.3390/microorganisms13092213

**Published:** 2025-09-22

**Authors:** Kun Dai, Lingli Ding, Xiaomeng Yang, Suqing Wang, Zhihui Rong

**Affiliations:** 1School of Nursing, Wuhan University, Wuhan 430071, China; 2024103070003@whu.edu.cn (K.D.); swang2099@whu.edu.cn (S.W.); 2Tongji Hospital Affiliated to Tongji Medical College, Huazhong University of Science and Technology, Wuhan 430030, China; dll200652@tjh.tjmu.edu.cn; 3School of Public Health, Wuhan University, Wuhan 430071, China; 2022203050022@whu.edu.cn

**Keywords:** preterm infants, gut microbiota, gut–brain axis, neurodevelopment, dysbiosis, SCFAs, tryptophan metabolism, probiotics

## Abstract

Preterm birth remains a significant global health challenge and is strongly associated with heightened risks of long-term neurodevelopmental impairments, including cognitive delays, behavioural disorders, and emotional dysregulation. In recent years, accumulating evidence has underscored the critical role of the gut microbiota in early brain development through the gut–brain axis. In preterm infants, microbial colonisation is frequently delayed or disrupted due to caesarean delivery, perinatal antibiotic exposure, formula feeding, and prolonged stays in neonatal intensive care units (NICUs), all of which contribute to gut dysbiosis during critical periods of neurodevelopment. This review synthesises current knowledge on the sources, temporal patterns, and determinants of gut microbiota colonisation in preterm infants. This review focuses on the gut bacteriome and uses faecal-sample bacteriome sequencing as its primary method of characterisation. We detail five mechanistic pathways that link microbial disturbances to adverse neurodevelopmental outcomes: immune activation and white matter injury, short-chain fatty acids (SCFAs)-mediated neuroprotection, tryptophan–serotonin metabolic signalling, hypothalamic–pituitary–adrenal (HPA) axis modulation, and the integrity of intestinal and blood–brain barriers (BBB). We also critically examine emerging microbiota-targeted interventions—including probiotics, prebiotics, human milk oligosaccharides (HMOs), antibiotic stewardship strategies, skin-to-skin contact (SSC), and faecal microbiota transplantation (FMT)—focusing on their mechanisms of action, translational potential, and associated ethical concerns. Finally, we identify key research gaps, including the scarcity of longitudinal studies, limited functional modelling, and the absence of standardised protocols across clinical settings. A comprehensive understanding of microbial–neurodevelopmental interactions may provide a foundation for the development of targeted, timing-sensitive, and ethically sound interventions aimed at improving neurodevelopmental outcomes in this vulnerable population.

## 1. Introduction

Preterm birth, defined as delivery before 37 weeks of gestation, remains a major global health challenge. In 2020, ~9.9% of births worldwide were classified as preterm, a rate that has improved little over the past decade and even reached 10.6% in 2014 [1,2]. Although advances in neonatal intensive care units (NICUs) have increased survival, long-term neurodevelopmental outcomes remain suboptimal—particularly in cognition, behaviour, and mental health [3]. Compared with term-born peers, preterm infants have higher risks of autism spectrum disorder (ASD), attention-deficit/hyperactivity disorder (ADHD), mood disturbances, and internalising symptoms, as well as motor and sensory dysfunctions (e.g., vestibular imbalance, altered pain perception, hearing loss, and features of cerebral palsy) [4,5,6]. Delays in language, cognition, sensory processing, and motor skills are frequent and often translate into poorer academic performance [7,8]. These adverse outcomes impose substantial psychological and economic burdens on families and pose persistent challenges to public health systems [9,10,11].

A growing body of research links preterm birth with altered neurodevelopment from infancy through adolescence, with potential long-term effects on brain structural connectivity [12,13]. The period between 20 and 40 gestational weeks is a critical “neurodevelopmental window” marked by rapid cortical expansion, massive neuronal migration, accelerated synaptogenesis, and the onset of myelination [14]: around 30 weeks, the subplate zone peaks in structural complexity and functional activity [15]. High-resolution foetal MRI indicates that significant white-matter tracts (e.g., cortico-thalamic pathways and corpus callosum) develop rapidly and non-linearly in utero, a trajectory complex to replicate ex utero [16]. Consequently, being born before the completion of microstructural remodelling and myelination is associated with persistent cortical immaturity, retention of subplate and early synaptic networks, and delayed myelination of primary fibres—neurobiological substrates for later deficits in sensation, cognition, and behaviour [14,17,18].

In parallel, late gestation and the early postnatal period are key windows for establishing gut microbiota, which shape brain development via the gut–brain axis. Disruptions in maternal microbial profiles may reprogram foetal neurodevelopment and affect postnatal behaviours [19,20,21,22]. Microbial-derived components—such as short-chain fatty acids (SCFAs), immune mediators (e.g., cytokines), enteroendocrine signals, and neuroendocrine circuits, including the vagus nerve and hypothalamic–pituitary–adrenal axis—act in concert to influence brain maturation and plasticity [23,24]. Animal studies show that these signals regulate gene expression, neurochemistry, and behaviours, including anxiety, memory, and motor function, and have been implicated in neuropsychiatric conditions such as depression, anxiety, and Parkinson’s disease [20].

Preterm delivery, by removing the foetus from the intrauterine environment before full maturity, simultaneously disrupts central nervous system development and early microbiota–brain interactions, potentially compounding neurodevelopmental vulnerability. Elucidating how variations in gut microbiota colonisation influence brain development through the gut–brain axis is therefore critical to understanding pathophysiology and designing targeted, evidence-based interventions for this high-risk population. Unless otherwise stated, “gut microbiota” refers to the bacteriome (typically profiled from stool as a practical proxy for intestinal communities); other microbiome domains—virome, archaea and fungi—are noted where relevant but are outside the scope of this review.

This narrative review searched PubMed/MEDLINE, Embase, the Web of Science Core Collection, and the Cochrane Library; ClinicalTrials.gov was screened for ongoing or recent trials. The strategy combined keywords and controlled vocabulary (MeSH/Emtree): (preterm/premature/very-low-birth-weight [VLBW]) and (gut microbiota/microbiome/bacteriome) and (neurodevelopment/white matter injury [WMI]/periventricular leukomalacia [PVL]/ASD/ADHD/emotional disorders), plus mechanistic/intervention terms SCFA signalling; tryptophan–serotonin/kynurenine; vagus hypothalamic–pituitary–adrenal (HPA) axis; barrier integrity; probiotics; prebiotics; human milk oligosaccharides (HMOs); skin-to-skin contact (SSC); antibiotic stewardship; faecal microbiota transplantation (FMT). Inclusion criteria were English, peer-reviewed studies involving preterm/very preterm/very-low-birth-weight (VLBW) populations or neonatal/animal work directly relevant to preterm neurodevelopment, reporting microbiota–neurodevelopment associations, objective outcomes (e.g., MRI, Bayley scales), or intervention effects. Exclusions were editorials, abstract-only records, single case reports without mechanistic value, and studies lacking relevant outcomes. Evidence was synthesised thematically; no formal risk-of-bias assessment or meta-analysis was undertaken. For orientation, we provide a cross-phenotype framework in Table 1 (Section 4), a study-level evidence summary in Table 2 (at the end of Section 4), and intervention details in Table 3 (Section 5).

## 2. Patterns and Determinants of Gut Microbiota Colonisation in Preterm Infants

### 2.1. Intrauterine Microbial Transmission: Sources and Controversies

#### 2.1.1. Microbial Evidence in Placenta, Amniotic Fluid, and Cord Blood

Recent studies have challenged the long-standing “sterile womb” paradigm by detecting low-abundance, low-diversity microbial communities in term placentas, amniotic fluid, and cord blood. Aagaard et al. first used shotgun metagenomic sequencing to identify low-biomass microbiota dominated by Proteobacteria and Firmicutes in 320 healthy placentas, with phylogenetic similarity to maternal oral microbiota, suggesting haematogenous translocation [44]. Similarly, Fardini et al. detected DNA from oral commensals such as *Fusobacterium* and *Streptococcus* in the basal plate of placental villi, supporting a “blood–placenta” transmission route [45].

In amniotic fluid, DiGiulio et al. employed both molecular and culture-based techniques to identify multiple bacterial species in preterm pregnancies, with microbial diversity correlating with intrauterine inflammation [46]. Collado et al. further detected Proteobacteria and Firmicutes signals in amniotic fluid and cord blood of term pregnancies [47]. Other studies reported anaerobic bacterial DNA, such as *Prevotella* and *Methylobacterium*, in human cord blood, implying potential transplacental passage into the foetal circulation [48].

Overall, low-biomass signals detected in placenta, amniotic fluid, and cord blood suggest episodic microbial DNA presence with phylogenetic ties to maternal oral sources, consistent with possible hematogenous transfer. However, the low abundance and contamination susceptibility of these samples warrant cautious interpretation.

#### 2.1.2. Meconium Microbiota and Maternal Influences

Meconium offers a valuable window into potential prenatal microbial exposure. Jiménez et al. successfully isolated cultivable strains of *Lactobacillus*, *Staphylococcus*, and Enterobacteriaceae from the meconium of 21 healthy term neonates, and confirmed their community structure through molecular methods [49]. In preterm infants, Mshvildadze et al. detected microbial DNA in 91% of meconium samples from 22 to 32 weeks’ gestational age (GA) using non-culture-based methods, with diversity positively correlated with gestational age [50]. Hu et al. reported that maternal diabetes significantly altered meconium microbiota, with increased Proteobacteria and decreased *Bacteroidetes* [51], highlighting gestational age and maternal metabolic status as key determinants of meconium colonisation.

Collectively, meconium findings indicate that putative prenatal and very-early postnatal microbial exposures are detectable and shaped by gestational age and maternal metabolic status. Yet the timing and persistence of these signals relative to actual foetal colonisation remain unresolved.

#### 2.1.3. Hypothesised Mechanisms of Maternal–Foetal Microbial Transmission

Multiple mechanisms have been postulated to explain maternal–foetal microbial transmission. In a murine model, Jiménez et al. administered GFP-tagged *Enterococcus* faecium to pregnant mice and later recovered genetically identical strains from the amniotic fluid and meconium of pups delivered by caesarean section, supporting a haematogenous route of transfer [52]. Alternative hypotheses propose that maternal phagocytic cells—such as macrophages and dendritic cells—may engulf gut bacteria and migrate across the placental barrier. Additionally, periodontal pathogens, including *Fusobacterium nucleatum* and *Porphyromonas gingivalis*, commonly associated with oral infections, have been observed to enter the maternal circulation and potentially access the intra-amniotic environment [53]. Nevertheless, these proposed pathways require further verification through direct microbial tracing studies.

In sum, multiple routes—hematogenous spread, trafficking of phagocytic cells, and translocation of oral pathogens—are biologically plausible and supported by animal tracing. Direct, unequivocal demonstration in humans, however, is still lacking.

#### 2.1.4. Controversies Surrounding in Utero Colonisation

Despite accumulating supportive evidence, the in utero colonisation hypothesis remains controversial. Lauder et al. found no significant differences between placental samples and laboratory contaminants under stringent negative controls, questioning the biological relevance of prior findings [54]. Dos Santos and Kennedy et al. similarly failed to detect specific microbial lineages in meconium compared to control samples, opposing the notion of intrauterine colonisation [55,56]. These findings underscore the importance of rigorous contamination control and suggest that current evidence remains inconclusive.

Taken together, current evidence from low-biomass samples is inconsistent and sensitive to the effectiveness of contamination controls. While in utero microbial exposure remains plausible, robust proof of persistent foetal colonisation is lacking.

#### 2.1.5. Bacterial Extracellular Vesicles (bEVs) and Prenatal Exposure

Accumulating evidence indicates that bacteria release nanoscale extracellular vesicles carrying lipids, proteins, and nucleic acids that can traverse epithelial and even the blood–brain barrier (BBB), thereby triggering and modulating host immune responses in the absence of culturable live bacteria [57,58]. Several studies further propose that maternally derived bEVs can enter the circulation via the oral or intestinal mucosa and reach the intrauterine environment, enter amniotic fluid, and “precondition” foetal immunity, thereby shaping neonatal gut colonisation and early immunoprogramming at birth [58]. In parallel, bEVs contribute to gut homeostasis through multiple mechanisms—including reinforcement of epithelial barrier function and orchestration of mucosal immunity—supporting a conceptual framework that emphasises “functional signal transmission” rather than “true colonisation” [59,60].

Collectively, bEVs constitute a biologically plausible conduit for prenatal microbial signalling in the absence of durable foetal colonisation. Yet current human evidence is inferential; causal claims will require harmonised isolation methods with rigorous contamination controls and longitudinal linkage of vesicle cargo to functional neonatal endpoints.

### 2.2. Temporal Succession of Gut Microbiota

Due to immature gut barrier, immune, and nervous systems—and frequent exposure to perinatal antibiotics—preterm infants display delayed and altered microbial colonisation patterns compared to term infants. During the first postnatal week, their faecal microbiota is dominated by low-diversity facultative anaerobes such as Enterobacteriaceae, *Enterococcus*, and *Staphylococcus*, while beneficial obligate anaerobes like *Bifidobacterium* and *Bacteroides* are markedly underrepresented [61,62,63].

With increasing postmenstrual age (PMA), the gut enters a “secondary succession” phase. Facultative anaerobes are gradually replaced by *Bifidobacteria* and Clostridia, with diversity increasing significantly by around 30 weeks PMA. Nonetheless, microbial maturity remains lagging behind that of term infants [64]. By 3–6 months post-weaning, most preterm infants partially achieve an adult-like microbiota composition, but overall community stability remains reduced [61,65]. In most cohorts, obligate anaerobes increase only after ~30 weeks PMA, and overall community stability often trails term infants by 2–3 months post-discharge [66].

Caesarean delivery and NICU-related exposures further delay the progression of gut microbial succession. Preterm infants delivered by caesarean section show reduced microbial diversity over the first six months of life, an elevated Proteobacteria-to-Firmicutes ratio, and persistently low abundances of *Bifidobacterium* and *Bacteroides*. NICUs’ colonisation is often dominated by opportunistic pathogens such as Enterobacteriaceae and *Staphylococcus*, with microbiota diversity showing substantial recovery only after six months [67,68].

Functionally, preterm stool samples exhibit reduced gene abundance for SCFA synthesis and enrichment in pathways related to xenobiotic metabolism, lipid degradation, and vitamin biosynthesis, suggesting immature microbial functionality. These deficits may compromise mucosal immunity and energy regulation, increasing the risk of necrotising enterocolitis (NEC), sepsis, and long-term neuroimmune-metabolic disturbances [63,64].

### 2.3. Key Influencing Factors

#### 2.3.1. Impact of Delivery Mode

The mode of delivery significantly affects the initial microbial colonisation of the gut and subsequent immune and metabolic development. During vaginal birth, the infant’s stomach is seeded with maternal vaginal and intestinal microbes, including *Lactobacillus*, *Bacteroides*, *Prevotella*, and *Sneathia*. These bacteria deplete oxygen to create an anaerobic environment conducive to *Bifidobacterium* proliferation and promote gut–immune axis maturation through metabolites such as SCFAs [69]. In contrast, caesarean delivery bypasses maternal birth canal exposure. Instead, the infant acquires microbes from the skin and hospital environment, including *Staphylococcus*, *Corynebacterium*, *Cutibacterium acnes (C. acnes)*, and *Enterococcus*, resulting in an approximately 30% reduction in key maternal-derived gut taxa such as *Bifidobacterium* and *Bacteroides* [70,71].

This “birth mode effect” contributes to persistent dysbiosis, with reduced diversity, skewed Firmicutes/Proteobacteria ratios, and higher risks of immune and metabolic disorders. C-section infants are more prone to asthma and allergic diseases, associated with diminished early microbial diversity and increased colonisation by skin-derived *Staphylococcus* aureus [72]. Longitudinal studies have also linked caesarean delivery to increased risk of childhood overweight and obesity, possibly due to insufficient *Bifidobacteria* and early microbial dysregulation of host energy metabolism [70].

#### 2.3.2. NICUs Environment and Antibiotic Exposure

The NICUs environment and empirical use of broad-spectrum antibiotics profoundly shape early microbiota development in preterm infants. NICUs often harbour nosocomial organisms such as *Klebsiella* and *Enterococcus*, which readily colonise the immature gut [73]. Broad-spectrum antibiotics further reduce microbial diversity, with overgrowth of Proteobacteria taxa including *Klebsiella* and Enterobacteriaceae, and depletion of *Bifidobacteria* and *Bacteroides* [43,67].

This dual impact of environmental exposure and antibiotic pressure disrupts microbial succession, delaying the establishment of a healthy, diverse community. It also correlates with increased risks of NEC and late-onset sepsis (LOS) [43,74]. Functionally, antibiotic-driven dysbiosis reduces SCFA production, impairs mucosal barrier integrity, and promotes the enrichment of antibiotic-resistant strains, thereby further increasing susceptibility to infection and inflammation [75,76]. While microbial diversity and beneficial taxa may partially recover within weeks of antibiotic cessation, substantial structural differences persist until 3–6 months of age [77,78].

Thus, optimising antibiotic stewardship in the NICUs and integrating targeted microbial interventions may help prevent severe complications and support the healthy development of the neuroimmune axis in preterm infants.

#### 2.3.3. Feeding Mode

##### Microbiota Benefits and Functional Roles of Human Milk

Breastfeeding plays a pivotal role in promoting the colonisation of beneficial gut bacteria in preterm infants, particularly strains of *Bifidobacterium* and *Lactobacillus*. This advantage is primarily attributed to the abundance of HMOs, which cannot be digested by the infant but serve as fermentable substrates for these probiotics [79]. Upon fermentation, these microbes generate SCFAs, such as acetate and propionate, which contribute to strengthening intestinal barrier function and promoting the differentiation of regulatory immune cells. These processes help suppress inflammation and reduce the likelihood of allergic diseases [80]. Furthermore, human milk contains secretory IgA, maternal immune components, and live bacteria that aid in mucosal colonisation and foster local immune tolerance. Collectively, these bioactive constituents facilitate the coordinated development of gut–brain axis signalling and neuroimmune regulation in early life [81].

##### Microbial Characteristics and Health Risks of Formula Feeding

Preterm infants fed with formula typically exhibit higher gut microbial diversity compared to exclusively breastfed infants; however, their dominant taxa tend to shift toward Firmicutes and Proteobacteria, including *Staphylococcus*, *Klebsiella*, and Enterobacteriaceae. In contrast, beneficial genera such as *Bifidobacterium* and *Lactobacillus* are markedly reduced [82]. This “high-diversity but low-beneficial” profile may result in decreased SCFA production, impaired gut barrier function, and suboptimal immune modulation, contributing to an increased risk of allergic and metabolic diseases [83].

##### Transitional Features and Interventional Potential of Mixed Feeding

Mixed-fed infants, who receive both human milk and formula, display a transitional microbial pattern. The abundance of *Bifidobacteria* and *Lactobacillus* in their gut microbiota typically falls between that of exclusively breastfed and formula-fed groups. Correspondingly, SCFA levels and immune biomarkers also show intermediate profiles [81]. This “transitional community” offers a potential window for targeted microbiota-based interventions, such as HMO supplementation or probiotic administration, especially in the NICUs setting, to optimise the coordinated development of the gut and immune systems in preterm infants.

#### 2.3.4. Maternal and Host-Related Factors

##### Maternal Diet and Metabolic Status During Pregnancy

Maternal body mass index (BMI), gestational weight gain, and dietary patterns can reshape the maternal microbiome and influence vertical transmission to the foetus, thereby significantly affecting the infant’s gut colonisation. Infants born to overweight or obese mothers exhibit increased levels of Bacteroides and Staphylococcus, and decreased levels of Bifidobacterium and Akkermansia, mirroring alterations in maternal microbiota [81,84]. A high-fat, low-fibre diet during pregnancy enriches proinflammatory taxa such as *Collinsella* and *Sutterella*, correlating with maternal metabolic profiles. Animal models further suggest that such diets can induce long-term expansion of Clostridiales and systemic inflammation in offspring [85,86,87].

Preterm infants born to mothers with gestational diabetes exhibit reduced gut α-diversity and a disrupted Firmicutes/Proteobacteria ratio [88].

##### Host Genotype and Microbial Regulation

Host genetic background indirectly shapes early gut microbiota by modulating immune responses, mucosal barrier integrity, and metabolic functions [89,90]. Comparative studies have found higher relative abundances of *Prevotella* and *Bifidobacteria* in the faeces of African rural children, while European urban children are enriched in *Bacteroides* [91,92]. Similar differences are observed between children living in urban slums in Bangladesh and suburban areas in the United States [93]. The INFABIO multicenter study highlighted that even after the introduction of complementary foods, gut microbiota assembly remains influenced by ancestral diet and genetic background [94]. In a Danish cohort, children with siblings had significantly higher microbial diversity, suggesting a role for familial genetic and microbial transmission in shaping microbiota maturation [75]. Studies in term infants also indicate that individual variability in community structure may be partially explained by TLR and MHC gene polymorphisms, suggesting a similar regulatory role in preterm infants. However, this hypothesis requires further validation in preterm-specific cohorts [95].

##### Family Structure and Geographic Environment

Family structure significantly influences gut microbiota development via shared microbial exposures. The “sibling effect” has been shown to accelerate gut microbial maturation: infants with siblings exhibit higher *Bifidobacterium* abundance and overall microbial diversity compared to only children, while those without siblings are more likely to harbour *Clostridium* and facultative anaerobes, with a reduced anaerobe/facultative anaerobe ratio [71,94]. Geographic factors also play a pivotal role in shaping initial colonisation. Children living in sub-Saharan African rural areas exhibit higher relative abundances of *Prevotella* and *Bifidobacterium*, whereas *Bacteroides* dominates European urban children. Similarly, Bangladeshi urban slum children show markedly different *Prevotella*/*Bacteroides* ratios compared to their U.S. suburban counterparts [91,92,96]. The INFABIO study further confirmed that both geographic location and genetic background continue to influence microbiota trajectories after weaning [97]. For preterm infants, exposure to different microenvironments due to hospital transfers or NICUs admissions across regions may result in distinct colonisation patterns, warranting systematic investigation.

In summary, the colonisation of gut microbiota in preterm infants is influenced by multiple interrelated factors, including mode of delivery, NICUs environment and antibiotic exposure, feeding practices, maternal metabolic status and dietary patterns, host genetic background, and geographical and family atmosphere. These factors interact across prenatal and postnatal stages, collectively shaping the early establishment and developmental trajectory of the gut microbial ecosystem, as illustrated in Figure 1.

## 3. Key Mechanistic Pathways of the Gut–Brain Axis in Neurodevelopment

### 3.1. Immune Pathways and White Matter Injury (WMI)

Dysbiosis in preterm infants can activate TLR-dependent proinflammatory cascades (notably TLR4–MyD88–NF-κB), elevating IL-1β/IL-6/TNF-α, disrupting the BBB, priming microglia, and injuring pre-oligodendrocytes—thereby impairing myelination and exacerbating WMI [98,99]. Germ-free models demonstrate that the absence of microbiota skews microglial maturation toward inflammatory states [100], while TNF-α can drive astrocyte-mediated pre-oligodendrocyte (pre-OL) apoptosis and demyelination [101]. Dysbiosis-biassed M1 microglial polarisation and early microbiota–immune axis disruption have been linked to neuroinflammation, circuit instability, and periventricular leukomalacia (PVL) in the preterm brain [102,103].

### 3.2. SCFA Pathways in Cognitive and Behavioural Regulation

SCFAs (acetate, propionate, butyrate) are key microbiota-derived signals that inhibit HDACs, promote oligodendrocyte differentiation/myelination, preserve BBB integrity, and enhance neurotrophins (e.g., BDNF), collectively mitigating preterm-related brain injury [104,105,106]. In vivo, butyrate augments neurogenesis and BDNF–TrkB signalling, improving cognition after hypoxic injury [106,107]. Germ-free and antibiotic-perturbation models demonstrate that microbiota and their metabolites facilitate microglial maturation [100,107,108], and microbial metabolites modulate astrocytic AhR to mitigate neuroinflammation [109]. Behaviourally, SCFAs can reverse stress-induced prefrontal and affective alterations, and probiotics/SCFAs ameliorate social and emotional phenotypes; SCFA-driven Treg induction further supports neuroimmune homeostasis [110,111,112].

### 3.3. Tryptophan–5-HT Pathways and Emotional Regulation

The microbiota shapes tryptophan fate along three routes with distinct neural consequences: serotonin synthesis via TPH (mood/emotion), kynurenine via IDO (neurotoxicity), and indole derivatives that engage AhR to regulate glial activity and neuroinflammation [113]. In demyelinating models, microbial indole metabolites activate astrocytic/microglial AhR, suppress NF-κB/STAT1, lower IL-6, and lessen demyelination, underscoring a microbiota–glia–emotion axis relevant to preterm neurodevelopment [114].

### 3.4. HPA Axis and Vagal Pathways in Learning and Memory

Early colonisation calibrates stress circuitry: germ-free mice exhibit exaggerated HPA responses that normalise with conventional/probiotic colonisation [21,115]. Microbial signals (metabolites, cytokines, neurotransmitters) engage hypothalamic CRH neurons to drive ACTH release and maintain gut–brain homeostasis [116]. Early-life stress perturbs serotonergic tone and 5-HT receptor expression across multiple regions [117,118]. Vagal afferents convey gut-derived 5-HT, GLP-1, and SCFAs to limbic and hypothalamic circuits, modulating cognition, metabolism, and memory; human functional magnetic resonance imaging (fMRI) demonstrates vagally mediated hippocampal responses to gastric/nutrient cues, while vagus-nerve stimulation enhances hippocampal plasticity and memory via BDNF [119,120,121,122,123,124].

### 3.5. Barrier Dysfunction and Neuroinflammation

Immature intestinal and neural barriers in preterm infants heighten permeability. Commensals upregulate epithelial tight-junction proteins to limit systemic translocation [125,126]. Microbiota and SCFAs likewise fortify the BBB by increasing junctional components [105], and neonatal probiotics (e.g., *Bifidobacterium infantis*) reduce IL-6/TNF-α and attenuate brain injury [22]. Conversely, barrier failure permits LPS entry, activating microglial TLR4 and propagating neuroinflammation [100].

## 4. Associations Between Gut Microbiota and Common Neurodevelopmental Disorders in Preterm Infants

As an overview, Table 1 summarises directional taxa, functional readouts, biological plausibility, and key confounders across phenotypes (Section 4.1, Section 4.2, Section 4.3 and Section 4.4).

### 4.1. WMI

WMI is one of the most prevalent forms of brain injury in preterm infants. It is widely recognised as a leading cause of cerebral palsy and neurocognitive impairment [127]. Based on MRI findings, WMI can be categorised into haemorrhagic infarction, PVL, diffuse WMI, and punctate white matter lesions (PWML) [6,127,128]. Among these, PVL is the most representative pathology, occurring in approximately 5% of extremely low birth weight (ELBW)infants and often accompanied by motor, cognitive, and visual impairments [129,130].

Recent studies suggest that, in addition to hypoxia–ischemia, the initiation of inflammatory and immune signalling cascades is a major contributor to the development of white matter injury (WMI) [129,131]. Increasing evidence indicates that gut microbial dysbiosis may influence WMI pathophysiology by modulating systemic inflammation, immune activation, and neuroimmune pathways through microbial-derived metabolites [132,133]. Specifically, decreased abundance of *Bifidobacteria* alongside overgrowth of *Clostridium* and Enterobacteriaceae members has been linked to a heightened susceptibility to WMI among premature neonates [134].

Microbial metabolites such as SCFAs are key regulators of microglial development and immune homeostasis. Erny et al. found that microglial development was impaired in germ-free mice and could be partially restored with SCFA supplementation, highlighting their role in central immune regulation [100]. Furthermore, the LPS–TLR4 signalling pathway is implicated in WMI: elevated TLR4 expression and expansion of LPS-producing bacteria (e.g., Enterobacteriaceae) in preterm infants may lead to pre-oligodendrocyte apoptosis via microglial activation [98]. Conversely, *Lactobacillus* and *Bifidobacterium* species may attenuate inflammation through enhanced SCFA production and modulation of TLR signalling, thereby reducing WMI risk.

Clinically, preterm infants with Grade III–IV intraventricular haemorrhage (IVH) are at greater risk for cystic PVL and severe motor impairments [135]. Even infants with Grade I PVL may experience persistent cognitive and emotional difficulties, suggesting long-term cortical connectivity alterations associated with early WMI [130,136].

### 4.2. ASD

Preterm birth significantly increases the risk of neurodevelopmental disorders, particularly ASD, characterised by social deficits, repetitive behaviours, and sensory abnormalities. Meta-analyses confirm that the risk of ASD is higher in preterm infants and correlates with lower gestational age, birth weight, and perinatal complications such as PVL [4,137,138].

Gastrointestinal dysfunction is highly prevalent in children with ASD (23–70%), including symptoms such as constipation, diarrhoea, and bloating, suggesting a role for gut microbiota in the pathophysiology of ASD [139,140,141]. Numerous studies have reported distinct microbial signatures in ASD children compared to typically developing (TD) peers, with reductions in beneficial genera (*Bifidobacterium*, *Prevotella*) and increases in potential pathobionts (*Clostridium*, *Desulfovibrio*, *Ruminococcus*) [142,143,144]. Notably, Clostridia produce neurotoxins and high levels of SCFAs like propionate, which are implicated in repetitive behaviours and social impairment [145,146].

A study comparing preterm children with and without ASD found higher α-diversity in the ASD group, suggesting a more heterogeneous but potentially pathogenic microbiota [28]. LEfSe analysis revealed increased abundance of Firmicutes, Clostridiales, *Ruminococcus gnavus*, and *Bifidobacterium longum*, alongside a depletion of anti-inflammatory species such as *Mitsuokella* and *Sutterella wadsworthensis*, which may underlie both gastrointestinal symptoms and social dysfunction [29].

At the metabolic level, faecal concentrations of propionate, phenylpropionate, and p-cresol are elevated in ASD children with ASD and correlate positively with behavioural and emotional severity [30,146]. In high-risk infants (e.g., siblings of ASD patients), reductions in Bifidobacteria, increases in *Clostridium*, and decreased GABA levels at 5 months predict poorer expressive language development [31].

Perinatal factors influencing microbial colonisation—including caesarean section, lack of breastfeeding, and antibiotic exposure—are associated with reduced *Bifidobacterium* colonisation and increased ASD risk [37,38]. Preliminary studies suggest that FMT and dietary interventions may alleviate core ASD symptoms and gastrointestinal issues by modulating gut microbiota composition [147,148].

### 4.3. ADHD

Associations in ADHD and emotional disorders are based on small, heterogeneous cohorts and should be considered hypothesis-generating [149]. ADHD, affecting approximately 5.3–7.2% of children globally, is characterised by inattention, impulsivity, and hyperactivity, often accompanied by language delays, social difficulties, and psychiatric comorbidities [150,151,152]. While its aetiology is multifactorial, disruptions in gut–brain axis function have been increasingly implicated [153].

Recent studies demonstrate that the gut microbiota composition in children with ADHD differs from that of neurotypical peers. A Dutch study found increased levels of *Bifidobacterium*, with associated genes involved in tyrosine metabolism, a precursor of dopamine [32]. In contrast, studies from China and Taiwan reported reduced abundance of beneficial genera such as *Faecalibacterium*, *Sutterella*, and *Dialister* [33,34,35], which may impair SCFA synthesis and neurotransmitter regulation.

SCFAs—particularly propionate, acetate, and butyrate—not only maintain gut barrier integrity but also modulate central serotonin (5-HT) and dopamine levels [154]. Lower SCFA levels have been observed in children with ADHD, with reductions in propionate and acetate correlating negatively with symptom severity [36].

Microbial diversity and stability during infancy are critical for cognitive development. A prospective study found that higher α-diversity at age one was associated with lower early learning scores at age 2, along with altered grey matter volume in the precentral gyrus and amygdala [26]. Deficits in expressive language commonly seen in ADHD may reflect early microbial disruption of GABA and 5-HT pathways [155].

Animal models support a causal role: Tengeler et al. demonstrated that FMT from ADHD patients into germ-free mice induced hyperactivity and anxiety-like behaviours, alongside altered tryptophan–serotonin metabolism [156]. Taxonomic analyses revealed increased *Bacteroides uniformis* and decreased *Faecalibacterium prausnitzii* in ADHD children, affecting SCFA production, inflammation, and neural signalling [34,157]. Additionally, lower plasma TNF-α levels and reduced microbial diversity in ADHD children suggest an immune–microbiota interaction in pathogenesis [158].

### 4.4. Emotional Disorders

Findings relating gut microbiota to emotional disorders in those born preterm are likewise preliminary; current associations stem from small, heterogeneous cohorts and should be considered hypothesis-generating.

Emotional disorders—primarily anxiety and depression—are prevalent among adolescents and adults born preterm. Epidemiological data indicate that extremely preterm birth (<28 weeks) and extremely low birth weight (<1000 g) are strong risk factors for later emotional problems [159]. In recent years, gut microbiota has emerged as a critical environmental factor influencing emotional development in this population.

In general paediatric populations, several studies have linked gut microbiota composition with emotional traits. For example, Jiang et al. reported decreased diversity and lower abundance of beneficial microbes in adults with depression [160]. Aatsinki et al. found that early microbiota features in 301 infants were associated with irritability, avoidance, and fussiness at 8 months of age [25]. Carlson et al. showed that infant microbial diversity and stability could predict social and emotional behaviour at age 2 [26]. In preterm infants, He et al. reported that dysbiosis during the first 6 months—characterised by reduced Bifidobacteria—was associated with maternal reports of anxiety and depressive symptoms at age 2 [27].

Animal studies provide further support. Zheng et al. demonstrated that faecal transplantation from depressed patients induced social withdrawal and depressive-like behaviours in germ-free mice [161]. Similarly, Kelly et al. found that microbiota from chronically stressed rats transferred depressive phenotypes to recipient animals [162].

Preterm infants often experience delayed colonisation and reduced microbial diversity due to antibiotic exposure, limited breastfeeding, and prolonged NICUs stays [163]. These colonisation patterns have been linked to elevated risk of later emotional disorders, warranting further investigation and targeted early interventions.

In summary, immune modulation, SCFA signalling, tryptophan–serotonin/kynurenine metabolism, HPA–vagal regulation and barrier integrity are the principal routes by which the gut–brain axis shapes preterm neurodevelopment; these interconnected pathways are illustrated in Figure 2.

## 5. Microbiota-Modulating Strategies and Clinical Interventions in Preterm Infants

### 5.1. Probiotics, Prebiotics, and Human Milk Oligosaccharides: Clinical Evidence and Potential

#### 5.1.1. Common Strains and Functional Components

Probiotics are viable microbes that confer health benefits when administered in adequate amounts [164]. In preterm infants, selected strains support colonisation and intestinal stability. They may offer neuroprotection through immune modulation, inflammation reduction, and effects on neurotransmitter pathways [165]—strains such as *Lactobacillus rhamnosus*, *L. plantarum*, *Bifidobacterium longum* subsp. *Infantis* (*B. infantis*), *B. breve*, and *L. reuteri* have demonstrated the capacity to reduce anxiety-related and depression-like symptoms in preclinical models [166,167].

Prebiotics—indigestible substrates that selectively stimulate beneficial taxa—include fructooligosaccharides (FOS) and galactooligosaccharides (GOS), which promote the growth of *Bifidobacterium* and modulate the activity of T cells, neutrophils, and dendritic cells [168]. HMOs, abundant in human milk, are fucosylated, non-fucosylated, or sialylated oligosaccharides [169] that act as natural prebiotics, lowering intestinal pH and increasing SCFAs [170,171]. HMO-adapted strains such as *B. infantis* are associated with more favourable white-matter development in preterm neonates [172,173].

Implementation notes. Candidate infants: very preterm or VLBW infants after establishment of minimal enteral feeds. Timing and duration: initiate early and continue until approximately 34–36 weeks’ PMA or until discharge. Dose and product quality: use hospital-grade preparations that provide ≥ 10^9^ CFU/day; strain-level labelling, batch traceability, and controlled storage are required [174]. Monitoring: record feeding intolerance, sepsis evaluations, and concomitant antibiotic exposures in a unit registry.

#### 5.1.2. Clinical Evidence and Target Populations

RCTs in term infants suggest cognitive and motor benefits with probiotics or HMO-enriched formulas [41,175], whereas evidence in preterm populations remains limited and heterogeneous [42]. Innovative approaches (e.g., maternal vaginal microbiota transfer, FMT) are being explored (SECFLOR, PREFLORE) [40,176]. The efficacy of prebiotics for neurodevelopment is uncertain and requires larger trials and mechanistic validation [98,177].

Translational perspective. Pragmatic use should be tailored to the feeding context (exclusive human milk versus fortified or formula feeding), individual variability (e.g., secretor status for human milk oligosaccharides), and institutional product quality assurance. Units should prespecify primary endpoints—NEC and LOS—and secondary neurodevelopmental outcomes (e.g., the Bayley Scales of Infant and Toddler Development) [178]. Key strains/doses and practical considerations are summarised in Table 3.

### 5.2. Antibiotic Stewardship and Skin-to-Skin Care in NICUs Settings

#### 5.2.1. Optimising Antibiotic Use and Preserving the Microbiota

Antibiotic exposure in NICUs is associated with reduced microbial diversity, delayed colonisation of beneficial taxa, expansion of resistant genes and opportunistic pathogens, and compromised intestinal and immune barriers, thereby increasing risks such as NEC [179,180]. Studies have shown that antibiotic-treated infants exhibit higher *Klebsiella* abundance and elevated expression of antibiotic resistance genes [181,182]. Notably, the occurrence of dysbiosis appears more closely linked to the presence of antibiotic exposure rather than its duration, underscoring the sensitivity of the microbiota to antimicrobial perturbations. Animal studies have further suggested that antibiotics may impair cognition by altering hippocampal neurogenesis and BDNF expression [183,184]. Therefore, narrow-spectrum antibiotics and shortened treatment durations are critical strategies. Recent RCTs highlight that rapid strain-level and resistome analysis can guide precise antibiotic selection while minimising microbiota disruption [182,185].

Implementation considerations. Initiate narrow-spectrum antibiotics only when clinically indicated; reassess at 48–72 h using culture and rapid resistome results to de-escalate or discontinue therapy; avoid prolonged empiric courses; and integrate stewardship checkpoints into feeding-advancement protocols to preserve early gut colonisation [186]. Stewardship checkpoints and typical de-escalation timelines are outlined in Table 3.

#### 5.2.2. Clinical and Long-Term Benefits of SSC

SSC, including kangaroo care, improves early gut colonisation and neurodevelopment by supporting gut barrier maturation and maternal microbial transfer; it modulates HPA axis activity, improves sleep architecture, and promotes prefrontal maturation [63,187], with associations to better neurobehavioural outcomes up to 10 years [39]. SSC also promotes breastfeeding and strengthens bonding, supporting neurodevelopment through microbiota-immune pathways and reducing maternal anxiety and depression [188,189,190].

Implementation considerations for SSC dosing: initiate as soon as the infant is clinically stable; set a daily cumulative-duration target and document minutes/hours as a reportable dose in the medical record; and embed SSC as a standard, recorded component of daily NICUs care with routine auditing [191,192]. Suggested SSC “dose” tracking and implementation tips appear in Table 3.

### 5.3. FMT: Frontiers and Ethical Considerations

#### 5.3.1. Emerging Applications and Target Populations

FMT has been shown to modulate neurodevelopment by enhancing SCFA production, maintaining microglial homeostasis, and reducing proinflammatory cytokine expression [100,193]. In germ-free mice colonised with stool from preterm infants, microbiota from donors of higher gestational age improved associative learning in adulthood [194]. Infant microbiota enriched in *Bifidobacteria* and *Bacteroides* is associated with more favourable cognitive phenotypes [195,196]. In ASD models, FMT improved behavioural outcomes and enhanced social interaction [147,197].

#### 5.3.2. Safety, Consent, and Ethical Implementation

For pregnant women and preterm neonates—vulnerable populations—microbiome-based interventions (probiotics, vaginal seeding, faecal microbiota transplantation, FMT) should occur under heightened protection thresholds and standardised oversight to reduce adverse events and procedural heterogeneity [198,199,200]. Ethical safeguards include prospective registration with transparent protocols to support reproducibility [201,202]; stratified eligibility by gestational age, delivery mode, and feeding type to reduce confounding [199,203,204]; predefined primary and safety endpoints with adverse-event reporting and long-term follow-up for rigorous evaluation [205,206]; establishment of standardised donor banks with resistome surveillance and end-to-end traceability to mitigate risks of pathogen transmission and antimicrobial resistance [204,205]; and multidisciplinary ethics review to ensure that regulation and clinical translation are grounded in the best available evidence [202]. Additional governance components include risk–benefit assessment with pre-specified stopping rules and independent data and safety monitoring board (DSMB) oversight, deferred consent with prompt countersignature in emergent NICUs contexts, stringent donor governance (pathogen/AMR screening, manufacturing/release compliance), attention to data governance and equity, and precise legal classification with staged lot release and post-marketing-style surveillance [199,200,207]. Regulatory exemplars include the FDA biologics pathway for FMT with adult recurrent *Clostridioides difficile* infection (rCDI) approvals—Rebyota (2022) and VOWST (2023) [208,209]—and, in the UK, NICE recommendations for FMT in rCDI with MHRA manufacturing requirements while FMT lies outside HTA oversight [210]. Animal data linking birth canal microbiota to neonatal health add biological plausibility to vaginal seeding [211].

NICUs priorities: stabilise early colonisation, protect the gut–brain axis, and avoid iatrogenic dysbiosis. Practice bundle: quality-assured, context-aligned probiotics; closed-loop antibiotic stewardship with 48–72 h review; SSC with a recorded daily dose; and FMT/vaginal seeding only under research/regulated pathways with donor governance and follow-up. See Table 3 for an at-a-glance checklist and implementation details.

## 6. Research Challenges and Future Directions

Despite growing interest in the gut–brain axis of preterm infants, clinical translation still faces multiple obstacles. Most studies continue to rely on 16S rRNA sequencing, which lacks sub-genus resolution and is susceptible to technical and sampling biases [212,213]. Although metagenomics and metabolomics are increasingly applied, protocols remain insufficiently standardised and are seldom embedded within longitudinal, mechanism-anchored frameworks. Environmental heterogeneity across NICUs (clinical settings, sampling windows, interventions) further drives inconsistent findings, and there is no consensus definition of a “high-risk” microbiome profile. Current animal models do not recapitulate disrupted intrauterine development or the complex, multi-exposure NICU milieu, underscoring the need for human-derived gut–brain organoids and co-culture systems. Knowledge gaps also persist for microbiota-targeted interventions—including optimal timing, dosing, and durability of effects across gestational ages and feeding contexts—whilst ethical and regulatory considerations, particularly for infant FMT, continue to evolve [214].

Looking ahead, progress will hinge on harmonised, longitudinal multi-omics cohorts with standardised sampling and data sharing; human gut–brain organoid platforms that better mirror early-life exposures; and rigorously designed clinical trials that prioritise neurodevelopmental outcomes. Trials should pre-specify standardised, developmentally meaningful endpoints and adopt objective, reproducible assessments, ensure adequate follow-up into early childhood, and include independent safety oversight. In parallel, development of a translational biomarker framework that integrates microbial, metabolic, and host inflammatory/barrier measures, together with mechanism-anchored trial designs and multi-centre collaboration under proportionate regulation, will be essential for responsible and effective clinical translation.

## Figures and Tables

**Figure 1 microorganisms-13-02213-f001:**
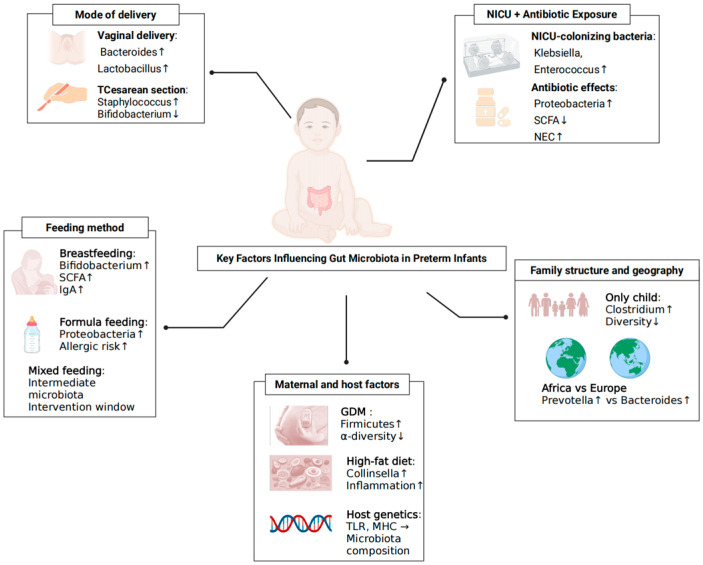
Key Factors Shaping Gut Bacteriome Colonisation in Preterm Infants. Determinants span delivery mode (vaginal vs. caesarean), NICUs environment and antibiotic exposure, feeding practices (human milk, mixed, formula), maternal factors (diet, BMI, gestational diabetes), host genotype/family structure and geography. Arrows indicate the typical direction of effects on early community succession; ↑ indicates an increase (higher abundance/risk); ↓ indicates a decrease (lower abundance/risk); → indicates a directional relationship.

**Figure 2 microorganisms-13-02213-f002:**
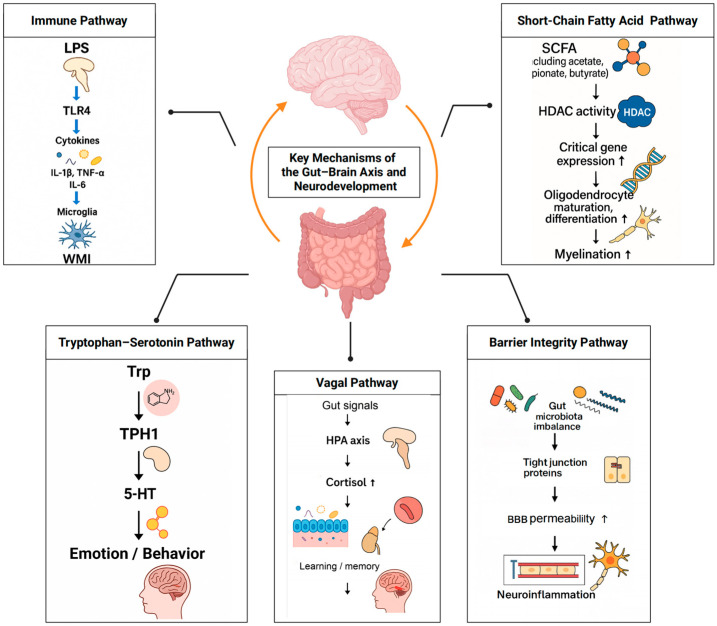
Core gut–brain axis pathways relevant to preterm neurodevelopment. Five interlinked pathways link the gut bacteriome to brain outcomes: (1) immune activation and microglial maturation; (2) SCFA-mediated myelination and BBB support; (3) tryptophan–serotonin/kynurenine-AhR–AhR signalling; (4) HPA axis and vagal pathways; (5) barrier integrity of gut and BBB. Representative mediators. Arrows indicate the direction of effect/flow within each pathway; ↑ denotes an increase.

**Table 1 microorganisms-13-02213-t001:** Microbiome features and functional readouts across neurodevelopmental phenotypes.

Phenotype	Directional Taxa *	Functional/Metabolic Signals	Biological Plausibility	Key Confounders	Mechanism Cross-Refs
WMI/PVL	Higher Enterobacteriaceae and *Clostridium*; lower *Bifidobacterium*	Higher LPS; lower SCFA production and availability	TLR4-mediated microglial activation leading to pre-OL injury; SCFA-supported myelination and BBB integrity	Infection, hypoxia–ischaemia, GA	Section 4.1
ASD	Lower *Bifidobacterium* and *Prevotella*; higher Clostridia; higher *Desulfovibrio*	Higher propionate and p-cresol; altered GABA and 5-HT signalling	SCFA- and neurotransmitter-mediated modulation	Delivery mode, antibiotic exposure, and feeding type	Section 4.2
ADHD	Lower *Faecalibacterium*, *Sutterella*, and *Dialister* (some studies report higher *Bifidobacterium*)	Reduced SCFAs; altered tryptophan–5-HT/DA pathway	Prefrontal network and arousal-system pathways	Diet, sleep, and comorbidities	Section 4.3
Emotional disorders	Lower α-diversity and depletion of beneficial taxa	Higher kynurenine; HPA-axis dysregulation	AhR–glial interactions and stress-axis links	Early adversity, maternal mental state	Section 4.4

* Directional taxa = taxa repeatedly reported as higher or lower in cases vs. comparators within neonatal/paediatric cohorts.

**Table 2 microorganisms-13-02213-t002:** Summary of human studies on gut microbiota and neurodevelopment in preterm cohorts: design, population, exposures/interventions, microbiome methods, outcomes, and level of evidence.

Domain	First Author (Year)	Design	Population	Exposure/Intervention	Microbiome Assessment	ND Outcome and Timing	Key Finding (Direction)	LoE
Emotional traits	Aatsinki (2019) [25]	Prospective cohort	Infants	Early microbiota features	Infant stool	Irritability/avoidance/fussiness at 8 months	Early microbiota associated with temperament traits	3
Social–emotional	Carlson (2018) [26]	Prospective cohort with MRI	Infants	Diversity and stability indices	Stool; MRI	Social and emotional behaviour at 2 years; grey-matter volume	Diversity/stability predicted social–emotional outcomes and brain structure	3
Preterm ND	Zhang (2024) [27]	Multicentre longitudinal cohort	Preterm infants	Early-life dysbiosis (Bifidobacteria)	Stool	Maternal-reported anxiety/depression at 2 years	Early dysbiosis is associated with higher emotional symptoms	3
ASD (preterm focus within cohorts)	Wan (2022) [28]	Case–control	Children with ASD vs. Tourette’s disorder	Microbiota composition/species markers	Stool	ASD diagnosis/behaviour	Underdeveloped microbiota; species markers distinguished ASD	3
ASD (classical)	Parracho (2005) [29]	Case–control	ASD vs. healthy	Composition	Stool	ASD	Taxa shifts (↑ Clostridia; ↓ beneficial taxa)	3
Cross-domain signals	Needham (2021) [30]	Cross-sectional	ASD vs. controls	Plasma/faecal metabolome	Metabolites (with or without stool)	Behavioural/emotional severity	↑ Propionate and ↑ p-cresol are associated with severity	3
ASD high-risk infants	Zuffa (2023) [31]	Prospective cohort (elevated-likelihood)	High-risk infants	Early microbiota and GABA	Stool; metabolites	Expressive language at 5 months and later	*Bifidobacteria*/*Clostridium* and GABA predicted poorer language	3
ADHD	Aarts (2017) [32]	Cross-sectional	Children with ADHD	Bifidobacteria	Stool: functional genes	Reward anticipation (neuroimaging/behaviour)	Microbiome features linked to dopamine precursor pathway	3
ADHD	Jiang (2018) [33]	Cross-sectional	ADHD (treatment-naïve) vs. controls	Composition	Stool	ADHD symptoms	Shifts in *Faecalibacterium*, *Sutterella*, *Dialister* vs. controls	3
ADHD	Wang (2020; 2022) [34,35]	Cross-sectional	Children with ADHD	Diet–microbiota patterns; psychostimulants	Stool; plasma cytokines	Symptom profiles	Taxa shifts; immune markers supported a microbiota–immune link	3
ADHD metabolites	Boonchooduang (2025) [36]	Cohort (pre/post medication)	Children with ADHD	SCFAs	Stool: SCFAs	Symptoms during treatment	SCFAs correlated with symptoms; drug effects on microbiota	3
Perinatal factors → colonisation	Azad (2016) [37]	Prospective cohort	Neonates/infants	Intrapartum antibiotics; birth mode; breastfeeding	Stool (Bifido/Lacto)	Colonisation dynamics	Caesarean/antibiotics linked to ↓ *Bifidobacterium* colonisation	3
Perinatal factors—Bifido development	Chen (2007) [38]	Cohort	Breast-fed neonates	Breastfeeding dynamics	Stool (Bifido/Lacto)	Early colonisation	Documented *Bifidobacteria*/*Lactobacilli* development with BF	3
SSC long-term ND	Bigelow & Power (2020) [39]	Longitudinal cohort	Infants	SSC dose	—	Neurobehaviour up to ~10 years	Better long-term neurobehaviour with higher SSC dose	2–3
VMT RCT	Zhou (2023) [40]	RCT (blinded)	Caesarean-delivered infants	Maternal vaginal microbiota transfer	Gut microbiome	Bayley/ND metrics	VMT improved microbiome; exploratory ND signals	2
Nutrition RCT (ND)	Colombo (2023) [41]	RCT	Term infants	MFGM + lactoferrin formula	—	ND at 9.5 years	Improved ND outcomes	2
Donor milk vs. formula	Colaizy (2024) [42]	RCT	Extremely preterm	Donor milk vs. preterm formula	—	ND	Group differences in ND (no microbiome measured)	2
Antibiotics → outcomes	Cotten (2009) [43]	Cohort	ELBW infants	Prolonged empiric antibiotics	—	NEC/death	Longer empiric therapy is associated with ↑ NEC/death	2–3

Abbreviations: ND = neurodevelopment(al); VMT = maternal vaginal microbiota transfer; MFGM = milk-fat globule membrane. LoE (Oxford CEBM): 1 = systematic review/high-quality RCT; 2 = RCT/strong cohort; 3 = cohort/case–control/cross-sectional. The arrow (→) indicates process flow/sequence (step to next step); ↑ indicates an increase (higher abundance/risk); ↓ indicates a decrease (lower abundance/risk).

**Table 3 microorganisms-13-02213-t003:** Microbiota-modulating strategies in the NICUs: mechanisms, timing, evidence, and safety and regulatory considerations.

Intervention	Target Population and Timing	Typical Preparation and Dose	Putative Mechanisms	Key Outcomes Reported	Evidence and Notes	Safety and Regulatory Considerations	Implementation Tips
Probiotics	Very preterm or VLBW; start after minimal enteral feeding; continue to ~34–36 weeks PMA or until discharge	Single- or multi-strain products (e.g., *Lactobacillus rhamnosus*, *L. plantarum*, *Bifidobacterium longum* subsp. *Infantis* (*B. infantis*), *B. breve*, *L. reuteri*); typically ≥ 10^9^ CFU/day; hospital-grade with strain-level labelling and batch traceability	Supports gut colonisation; reduces inflammation; modulates immune function; potential effects on neurotransmitter pathways	Reduced NEC, LOS, and mortality (effects heterogeneous across populations); mixed findings for cognitive and motor outcomes.	The overall evidence is mixed, with substantial heterogeneity in strains, product quality, dose, and start time.	Emphasise product quality and traceability; monitor adverse events; avoid in severe immunodeficiency or high-risk short-bowel contexts.	Unit SOPs specifying eligibility, initiation and discontinuation criteria, batch retention, and pharmacy oversight
Prebiotics (FOS/GOS)	Preterm infants during feeding-transition phases, particularly when formula-fed	Formula additives; dose per product specification; follow unit SOPs	Selective promotion of *Bifidobacterium*; immune modulation (T cells, neutrophils, dendritic cells).	Improved gut colonisation; neurodevelopmental outcomes remain uncertain.	Larger randomised trials and mechanistic validation needed; interpret separately from HMOs	Generally safe; monitor for bloating and feeding intolerance.	Pair with probiotic and human-milk strategies; consider gradual introduction.
HMOs	Prioritise human milk; consider HMO-supplemented formula when needed	Blends enriched in fucosylated and/or sialylated HMOs; align product choice with feeding context (exclusive human milk vs. fortified/formula)	Natural prebiotics; lower luminal pH; increase SCFAs; support immune homeostasis	Associations with favourable white-matter signals in preterm cohorts.	Evidence is predominantly observational and mechanistic; effects may vary by secretor status	Generally, a safe profile is acceptable; ensure regulatory compliance for HMO-containing products.	Strengthen lactation support; when exclusive human milk is not feasible, consider HMO-supplemented formula with tolerance monitoring.
Antibiotic stewardship	All NICUs’ anti-infective scenarios	Narrow-spectrum therapy first; culture-guided reassessment at 48–72 h; integrate rapid resistome results	Preserve microbial diversity, limit the expansion of antibiotic-resistant genes (ARGs), and protect epithelial and immune barriers.	Potential indirect reductions in NEC and LOS; animal data suggest cognition-related risks with dysbiosis.	A strong rationale for programme-based stewardship is required, which necessitates local resistance data.	Stewardship committee; access to rapid diagnostics	Closed-loop process: initiate when indicated → reassess at 48–72 h → deescalate or discontinue; link checkpoints to feeding-advancement protocols.
SSC/KC	Initiate as soon as clinically stable; set and record a cumulative daily duration as a reportable “dose”	Continuous or intermittent sessions; either parent can provide SSC	Maternal microbial transfer, support of gut-barrier maturation, HPA axis modulation, and improved sleep architecture and prefrontal maturation.	Maternal microbial transfer; gut-barrier maturation; HPA-axis modulation; improved sleep architecture and prefrontal maturation	Robust evidence base; low cost	Generally safe; ensure thermal stability and securement of lines/monitoring devices	Embed SSC as a charted, prescribable component of daily NICUs care; record cumulative minutes per day and audit adherence.
FMT and vaginal seeding	Research-only with strict indications; pregnant women and neonates are vulnerable populations.	Standardised donor screening and governance; GMP-compliant manufacturing; outside the NICUs, recurrent rCDI remains the primary clinical indication.	Increases SCFAs, supports microglial homeostasis, reduces proinflammatory cytokines; behavioural improvements in preclinical models reported.	Increases SCFAs; supports microglial homeostasis; reduces proinflammatory cytokines; behavioural improvements in preclinical models	High regulatory and ethical threshold; prospective registration and DSMB oversight required	Precise legal classification; staged lot release and recall procedures with post-release surveillance.	Conduct only within IRB-approved NICUs protocols; where permitted, consider deferred consent with prompt countersignature.

Abbreviations: GMP, Good Manufacturing Practice; IRB, Institutional Review Board; ARG, Antibiotic Resistance Gene(s). Notes: Doses/examples are illustrative and should conform to institutional protocols; the arrow (→) indicates process flow/sequence (step to next step). For evidence and detailed references, see Section 5.1, Section 5.2 and Section 5.3 of the manuscript.

## Data Availability

No new data were created or analyzed in this study. Data sharing is not applicable to this article.

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
