# Peer review of "Gut Microbiota and Neurodevelopment in Preterm Infants: Mechanistic Insights and Prospects for Clinical Translation"

_microorganisms, 2025, doi:10.3390/microorganisms13092213_

Round 1

Reviewer 1 Report

Comments and Suggestions for Authors

This is an interesting and well-written article that reviews the interaction of preterm birth (which is usually associated with caesarean delivery, perinatal antibiotic exposure, formula feeding, and stays in intensive care units) with gut bacterial colonisation and neurodevelopmental outcomes; it also discusses microbiota-targeted interventions and some ethical and legal issues related to these interventions. 

General comments

Please clarify that this study specifically focuses on the gut bacteriome, most probably the faecal bacteriome (as a proxy for the gut bacteriome), since the gut microbiota is complex and may include not only bacteria but also some viruses, archaea, and fungi. 

When discussing placental microbiota in Section 2, please be cautious, as it is still a topic of debate. While some researchers have reported low biomass levels in the placenta and amniotic fluid, other studies suggest that bacterial contamination might be the cause. These conflicting findings warrant further discussion.  

The authors could also discuss the role of bacterial extracellular vesicles to explain the presence of bacteria in the placenta and amniotic fluid, as well as their potential role in the colonisation of the infant’s gut (see, for example, https://doi.org/10.3390/ijms21238959; https://doi.org/10.1016/j.mam.2021.100955)

Section 5.3: Ethical considerations warrant further discussion, as both pregnant women and preterm infants are vulnerable groups requiring special care. Additionally, the mention of informed consent is noted but not elaborated upon. What ethical factors should be considered when legislating treatments and interventions? Ethical review systems and legislation for microbiota-based interventions in neonates should be based on the best science available.

Minor points

The figures require more detailed explanations. The summaries preceding each figure could serve as figure legends.

The writing combines British and American English. Either is acceptable, but it must be consistent.

Please verify the scientific formatting of bacterial names at the genus and species levels, specifically whether they should be italicised.

Line 319: Regulation

Lines 545-553: The paragraph is misplaced.

Author Response

Dear Reviewer,
We sincerely appreciate your careful review and valuable suggestions. Our detailed replies and the revised manuscript are provided in the attachment for your consideration. Thank you for helping us improve the paper.

Reviewer 2 Report

Comments and Suggestions for Authors

Reviewer Report

General Assessment

            The manuscript presents a comprehensive and timely review of the relationship between gut microbiota and neurodevelopment in preterm infants, with a focus on mechanistic pathways along the gut–brain axis and their clinical relevance. The topic is highly relevant, of translational importance, and fits well within the scope of Microorganisms. The manuscript is rich in references and up to date. Nevertheless, in its current form the text is overly extensive and occasionally repetitive, and it would benefit from a more balanced emphasis between mechanistic detail and clinical applicability.

Strengths

  • Topical and original of the theme is timely and of significant translational interest for neonatology and neuroscience.
  • Well-structured, with logical flow from microbial colonisation to mechanisms, clinical implications, and interventions is clear.
  • Extensive and recent bibliography, which are broad and include recent studies (2022–2023).
  • Multidimensional approach, from maternal, environmental and nutritional to clinical factors which are integrated.
  • The manuscript has an educational value and is useful both for clinicians and for early-career researchers.

Weaknesses

  • Excessive length - the manuscript is very long, which may reduce readability.
  • Imbalance - mechanistic details are over-emphasised compared with translational and clinical aspects.
  • Redundancy - certain concepts (e.g. SCFAs, HPA axis) are repeated across sections.
  • Limited visual synthesis - tables summarising clinical trials or interventions are lacking.
  • Stylistic clarity - some paragraphs contain overly long sentences that would benefit from simplification.

Suggestions for Improvement

  • Condense the introduction and mechanistic sections to improve clarity and flow.
  • Expand and strengthen the translational perspective by providing a more detailed overview of clinical interventions (probiotics, prebiotics, FMT, skin-to-skin contact), ideally summarised in tabular form.
  • Present controversial areas (e.g. intrauterine colonisation) in a more balanced manner, clearly stating the lack of consensus.
  • Add tables or figures that synthesise microbiota profiles associated with different neurodevelopmental disorders.
  • Simplify phrasing and remove redundancies for improved readability.
  • Conclude with a forward-looking section outlining specific future directions, such as biomarkers and clinical trial design.

Conclusion

            The manuscript is of high scientific value and could make a meaningful contribution to the literature. However, substantial revision is required in order to improve readability, balance, and translational depth.

Recommendation: Major Revision

Author Response

(The authors gave the same response as above.)

Reviewer 3 Report

Comments and Suggestions for Authors

This is a clear, timely, and valuable review. I appreciated the topic which was very interesting. The structure is logical and the translational angle is useful for clinicians. I recommend just little improvements: 

  1. Remove leftover editorial instructions (Section 5.1.1).
    Delete the sentence: “The introduction should briefly place the study in a broad context…”—it appears to be author guidelines rather than content_lines 545-552.

  2. Add a brief methods paragraph for transparency (after the Introduction) describing that it is a narrative review, which keywords were used for searching papers and how were included papers

  3. Close each intrauterine-colonization subsection with a 1–2 sentence synthesis (Sections 2.1.1–2.1.4).
    Example for 2.1.4:
    “Taken together, current evidence from low-biomass samples is inconsistent and sensitive to contamination controls. While in utero microbial exposure remains plausible, robust proof of persistent fetal colonization is lacking.”

  4. Clarify the maturation timeline in Section 2.2 (temporal succession).
    Add one sentence quantifying delay:
    “In most cohorts, obligate anaerobes increase only after ~30 weeks PMA, and overall community stability often trails term infants by 2–3 months post-discharge.”

  5. Flag evidence as preliminary in ADHD/emotional-disorders sections (4.3–4.4).
    Add an opening line:
    “Associations in ADHD and emotional disorders are based on small, heterogeneous cohorts and should be considered hypothesis-generating.”

  6. Add one summary table to improve readability with the included relevant studies, briefly describing the characteristics of them, including level of evidence

  7. Do you think it could be possible to add a short “Implications for practice” paragraph at the end of Section 5?

  8. Fix typographical/encoding artifacts and ensure acronym consistency.
    Replace “faÄ´y/paÄ´erns” with “fatty/patterns”; define NICU, SCFA, HPA, BBB, PVL, WMI at first use and use consistently thereafter.
  9. The intrauterine sections are citation-dense; add 1–2 integrative sentences to weigh the evidence. Conversely, consider adding recent cohort/RCT references (where available) to ADHD/emotional-disorder sections and to probiotics/HMOs in preterm infants.

Author Response

(The authors gave the same response as above.)

Round 2

Reviewer 1 Report

Comments and Suggestions for Authors

In this revised and improved version of the document, the authors have addressed all my comments, and the effort is appreciated. Just maintain consistency when spelling 'fetus' or 'foetus' and 'fetal' or 'foetal.' In lines 535-536, italics should be limited to the scientific name.

Reviewer 2 Report

Comments and Suggestions for Authors

The revised version of the manuscript meets the criteria for publication.